# Rethinking the CSC Model for Natural Images

**Dror Simon**
Department of Computer Science
Technion, Israel
dror.simon@cs.technion.ac.il

**Michael Elad**
Department of Computer Science
Technion, Israel
elad@cs.technion.ac.il

## Abstract

Sparse representation with respect to an overcomplete dictionary is often used when regularizing inverse problems in signal and image processing. In recent years, the Convolutional Sparse Coding (CSC) model, in which the dictionary consists of shift invariant filters, has gained renewed interest. While this model has been successfully used in some image processing problems, it still falls behind traditional patch-based methods on simple tasks such as denoising. In this work we provide new insights regarding the CSC model and its capability to represent natural images, and suggest a Bayesian connection between this model and its patch-based ancestor. Armed with these observations, we suggest a novel feed-forward network that follows an MMSE approximation process to the CSC model, using strided convolutions. The performance of this supervised architecture is shown to be on par with state of the art methods while using much fewer parameters.

## 1   Introduction

The field of image restoration deals with the recovery of degraded images. Popular forms of degradation include an additive noise, a blurring kernel, missing pixels, and more. Retrieving an image from its degraded version is typically an ill-posed problem. Therefore, to enable the inversion task, it is necessary to include prior information on the original signal. An image prior, also referred to as an image model, relates to a mathematical description of the image true distribution. In the past 2-3 decades, many such models have been suggested and deployed. Some of these include reliance on spatial smoothness, self-similarity, and sparse representation [1–4]. The later is the focus of this work.

The sparse representation model has been successfully incorporated in various signal and image processing applications [2, 5–9]. This model assumes that a signal $X \in \mathbb{R}^N$ is formed by a linear combination of only a few *atoms*, taken from the *dictionary* $D \in \mathbb{R}^{N \times M}$, i.e. $X = D\Gamma$, where $\Gamma \in \mathbb{R}^M$ is sparse. When a noisy signal $Y = X + V \in \mathbb{R}^N$ is at hand ($V$ is a bounded energy noise: $\|V\|_2 \leq \epsilon$), seeking for its sparse representation $\widehat{\Gamma}$, leads to an estimation of the original signal via $\widehat{X} = D\widehat{\Gamma}$. Finding $\widehat{\Gamma}$ is commonly referred to as *sparse-coding* or a *pursuit*, formulated as

$$\min_{\Gamma} \|\Gamma\|_0 \quad \text{s.t. } \|D\Gamma - Y\|_2 \leq \epsilon, \tag{1}$$

where the $\ell_0$ pseudo-norm counts the number of non-zeros in the vector.[1] Sparse coding is NP-hard in general [10], hence approximation methods are used. A common approach replaces the $\ell_0$ pseudo-norm with the $\ell_1$, leading to a convex problem termed Basis-Pursuit (BP) [11]. The BP method has been theoretically analyzed [12], shown to successfully recover a solution close to the original sparse representation, depending on properties of the dictionary and the cardinality of the sought solution.

The dictionary is an important ingredient in the formation of this prior, as its atoms characterize the signals that this model can represent sparsely. Learning the dictionary from the corrupted signal

itself, or from an external dataset, has been shown to be quite effective, leading to the development of various dictionary learning algorithms and their use [13–16]. Unfortunately, due to the curse of dimensionality, these algorithms are applicable only for reasonably sized signals. Many algorithms overcome this limitation by dividing the complete signal (e.g. a complete image) into fully overlapping small *patches*, treating each independently [2, 14, 15, 17]. This treatment consists of imposing the sparse representation model on the patches using a local dictionary $D_L \in \mathbb{R}^{n \times m}$,

$$\forall i : \quad \min_{\boldsymbol{\alpha}_i} \|\boldsymbol{\alpha}_i\|_0 \quad \text{s.t.} \ \|D_L \boldsymbol{\alpha}_i - P_i Y\|_2 \leq \epsilon, \tag{2}$$

where $P_i \in \mathbb{R}^{n \times N}$ extracts the $i$-th patch from $Y$ ($n \ll N$), and its representation $\boldsymbol{\alpha}_i \in \mathbb{R}^m$ is assumed to be sparse. Once clean estimates of the patches are found, this process proceeds by *Patch Averaging* (PA), i.e. merging all the refined patches together to form a final global estimate of the clean image:

$$\widehat{X} = \frac{1}{n} \sum_i P_i^T D_L \boldsymbol{\alpha}_i, \tag{3}$$

where $P_i^T$ places $D_L \boldsymbol{\alpha}_i$ in the $i$-th location in the constructed image. Intuitively, operating independently on patches must be sub-optimal, since the dependencies between the patches are falsely neglected [18]. To overcome this flaw, past work suggested enforcing the local prior on the patches of the merged image [17, 19], leveraging the self-similarity between different patches [20], and more.

Recently, there has been a renewed interest in global models that may overcome this local-global dichotomy. The Convolutional Sparse Coding (CSC) prior [21, 22] replaces the traditional patch-based model with a global shift-invariant one. Instead of operating on patches, it suggests a global dictionary constrained by a specific structure – a concatenation of banded circulant matrices[2], limiting the degrees of freedom introduced by the general sparsity-based model. Various algorithms have been suggested to efficiently handle the global pursuit [23–27]. These methods have been augmented by efficient dictionary learning algorithms [26, 28–30]. A recent work provided theoretical guarantees for the CSC model and its corresponding global pursuit results [31].

The CSC model has shown great success in several natural image processing tasks such as image separation, image fusion, and super-resolution, matching or outperforming local-based methods [26, 28, 32–34]. Interestingly, one can find two common properties to all these success stories. The first is the fact that the CSC is merely used as a complementary component, modeling only the texture part of the image, after stripping its low-frequencies. Second, these applications assume noiseless images, and thus the CSC cannot fail in over-fitting the data. Indeed, when brought to other classical tasks, such as image denoising or other inverse problems that involve an additive noise, the CSC has been shown to fail utterly.

The main contribution of this work is in providing novel insights regarding the CSC for modeling natural images, and extending the applicability of this prior while tying it to deep-learning. We propose an explanation for the incompetence of this model in representing natural images reliably, and show that PA can be perceived as a Minimum Mean Square Error (MMSE) approximation to the CSC. Building on this, we suggest to improve this approximation and obtain a CSC estimation process that operates directly on an image without any pre-processing steps. Finally, we leverage these observations to implement a feed-forward Convolutional Neural Network (CNN) whose layers strictly correspond to each step in the processing flow of sparse-coding based image denoising. Our results are on par with current state of the art supervised methods while drastically reducing the number of parameters.

## 2 Background: Convolutional Sparse Coding

### 2.1 The CSC Model

The CSC model considers a shift-invariant property in the signal, by assuming that[3] $X \in \mathbb{R}^N$ is constructed by a sum of $m$ convolutions of sparse feature maps $\{Z_i\}_{i=1}^m \in \mathbb{R}^N$ by filters $\{d_i\}_{i=1}^m$ of

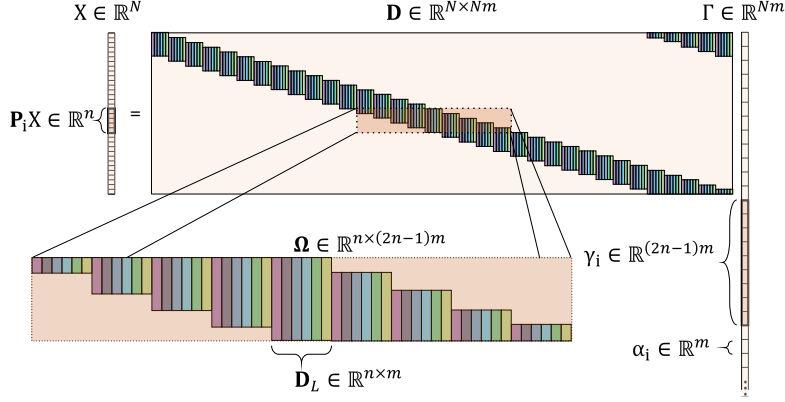

Figure 1: The CSC model and its components.

length $n \ll N$. CSC then refers to solving the following optimization problem:

$$\min_{\{\boldsymbol{Z}_i\}_{i=1}^m} \sum_{i=1}^m \|\boldsymbol{Z}_i\|_0 \quad \text{s.t.} \quad \boldsymbol{X} = \sum_{i=1}^m \boldsymbol{d}_i * \boldsymbol{Z}_i. \tag{4}$$

Equivalently, we define a single global sparse representation vector $\boldsymbol{\Gamma} \in \mathbb{R}^{Nm}$, constructed by interlacing the sparse feature maps $\{\boldsymbol{Z}_i\}_{i=1}^m$. A global dictionary is then composed as follows: Let $\boldsymbol{D}_L \in \mathbb{R}^{n \times m}$ represent a local dictionary whose columns are the filters $\{\boldsymbol{d}_i\}_{i=1}^m$; then $\boldsymbol{D}$ contains $N$ shifts of this local dictionary – see Figure 1. Under this description, (4) is equivalent to

$$\min_{\boldsymbol{\Gamma}} \|\boldsymbol{\Gamma}\|_0 \quad \text{s.t.} \quad \boldsymbol{X} = \boldsymbol{D}\boldsymbol{\Gamma}. \tag{5}$$

When noisy measurements $\boldsymbol{Y}$ are at hand, (5) is modified to allow for variations in the signal, leading to the problem defined in Equation (1).

We introduce additional definitions, taken from [31], which will aid in our exposition. The sparse representation vector $\boldsymbol{\Gamma}$ can be thought of $N$ concatenated vectors $\boldsymbol{\alpha}_i \in \mathbb{R}^m$, termed *needles*. Each describes the contribution of the $m$ filters when aligned to the $i$-th element in $\boldsymbol{X}$, i.e.

$$\boldsymbol{X} = \boldsymbol{D}\boldsymbol{\Gamma} = \sum_{i=1}^N \boldsymbol{P}_i^T \boldsymbol{D}_L \boldsymbol{\alpha}_i = \sum_{i=1}^N \boldsymbol{P}_i^T \boldsymbol{s}_i, \tag{6}$$

where we have defined the *slice* $\boldsymbol{s}_i = \boldsymbol{D}_L \boldsymbol{\alpha}_i$. Observe the resemblance between this and the patch-averaging in Equation (3). This may suggest that the CSC is in-fact a global model extending PA. If this was true, one could have expected the CSC to be at least as good as the local model on any image processing task. Is this the case? Keep reading.

Similarly, $\boldsymbol{P}_i \boldsymbol{X} = \boldsymbol{P}_i \boldsymbol{D}\boldsymbol{\Gamma}$ is a patch of size $n$ extracted from $\boldsymbol{X}$. Equivalently, one may write $\boldsymbol{P}_i \boldsymbol{D}\boldsymbol{\Gamma} = \boldsymbol{\Omega}\boldsymbol{\gamma}_i$, where a *stripe* $\boldsymbol{\gamma}_i$ concatenates $2n-1$ needles and $\boldsymbol{\Omega}$ is termed the *stripe dictionary*. Figure 1 demonstrates these definitions. An analysis of the convolutional sparse coding problem is proposed in [31], showing that when all the stripes $\boldsymbol{\gamma}_i$ are sparse,[4] the solution to (5) is unique. Moreover, under the same conditions, BP is guaranteed to retrieve this solution. An analysis was also given for the noisy case, showing that under similar conditions, the solution attained by BP is stable.

## 2.2 CSC in Practice

The first application we mention is cartoon-texture separation, where the goal is to blindly decompose an image into its texture and cartoon parts. Recent papers have achieved successful results by incorporating the CSC model [28, 34]. Curiously, these algorithms model the cartoon image via the Total-Variation smoothness assumption, while using the CSC to model only the texture.

A second application where the CSC achieves satisfactory results is image fusion [26, 33]. Here the goal is to integrate complementary information from multiple source images of the same scene.

The results obtained by integrating the CSC model surpass those achieved by patch-based methods on various metrics [33]. In such an algorithm, each image is first decomposed to smooth and detail layers. The fusion itself is obtained by computing the convolutional sparse representation of the detailed layers, and merging these by a pixel-wise max-pooling strategy.

Another application where CSC has been demonstrated to perform very well is single-image-super-resolution. Here, the objective is a high resolution (HR) image, obtained from a low resolution (LR) one. In [32] a CSC scheme is suggested, leading to superior results over patch-based methods. As in the image fusion task, the algorithm separates the LR image into a smooth and a residual image. While the smooth part is simply interpolated, the residual is coded using CSC, and the final details image is recovered by applying a set of filters on the obtained sparse representations.

In all these successful applications, the input image is first separated into smooth and non-smooth images. Then, without a formal reasoning, the CSC only models the detail-rich content of the image, hinting to its limitations. Why does the CSC perform well only on non-smooth signals? We answer this question in the following sections. Another common feature to these successful applications is the fact that the data is assumed to be noiseless. Is this a coincidence? Can the CSC be of benefit on noisy natural images? In order to answer these questions let us refer to an unsuccessful use of the CSC: image denoising. Applying the CSC model directly on the noisy image leads to disappointing results, falling far behind PA [25, 35, 36]. We emphasize that the same is true for other applications where noise cannot be neglected, such as deblurring and other inverse problems.

To date, no CSC denoising algorithm competes favorably with the PA method on natural images, with the exception of [37]. This work extends the concept of Learned Iterative Soft Thresholding (LISTA) [38–40] to CSC, unfolding the pursuit algorithm into a recurrent network. The results obtained are on par with the K-SVD algorithm [3].[5] The last part of our work is closely related to [37]. By adopting our insights on the CSC model and its MMSE approximation, we offer a CSC deployment that leads to enhanced denoising performance that are on par with the most recent supervised methods.

## 3 Why Does the CSC Model Denoise Natural Images Poorly?

### 3.1 Poor Coherence

The work in [31] has show that the theoretical uniqueness and stability guarantees for the convolutional sparse coding problem are conditioned on the maximum number of local non-zero elements in $\|\mathbf{\Gamma}\|$

$$\|\mathbf{\Gamma}\|_{0,\infty} < \frac{1}{2}\left(1 + \frac{1}{\mu(\mathbf{D})}\right),\tag{7}$$

where $\mu(\mathbf{D})$ is the mutual coherence of $\mathbf{D}$, i.e. the max absolute normalized cross-correlation between its columns.[6] Moreover, under the same conditions, BP is guaranteed to recover this solution. This bound implies that to allow for a large number of active filters in the signal while keeping the solution accessible, the filters and all their shifts must have low cross-correlations. Specifically, the auto-correlation of the filters should be low as well. Unfortunately, this property does not align with the characteristics of natural images. For the most part, these consist of piece-wise smooth regions and occasional textures. This structure is key in many denoising and compression methods [1, 41]. Hence, to allow for a sparse representation, a convolutional dictionary must contain smooth or piece-wise smooth filters. That said, the auto-correlation of a these filters decay slowly, leading to highly correlated atoms in the global dictionary, restricting the number of non-zero elements allowed in each stripe $\mathbf{\gamma}_i$ while satisfaying the above bound. For example, if a dictionary contains the constant (DC) filter, the maximum number of non-zeros allowed in a stripe to assure uniqueness is 1, enforcing unpainted pixels in the signal.

Generally, a CSC representation of a natural image imposes a contradiction between the cardinality of the sparse representation and the coherence of the global dictionary. To assure a sparse representation, the former requires piecewise smooth filters, whereas the latter demands low global mutual-coherence, which counters these slow-changing filters. In its current form, the CSC cannot satisfy the two demands simultaneously, making it unsuitable for natural images. Note that the successful applications

that were mentioned in Section 2.2, apply the CSC model only on the texture rich part of the image, leading to Gabor-like non-smooth filters, thus avoiding the described conflict.

## 3.2 A Bayesian Standpoint

From a Bayesian point of view, the solution to the problem posed in Equation (1) (or its Lagrangian form) corresponds to the Maximum A-posteriori Probability (MAP) estimator under a sparse prior [42–45]. Clearly, this solution is inferior to the Minimum MSE (MMSE) estimator in terms of MSE when the two differ. As we show next, as opposed to a convolutional pursuit (being MAP approximation), PA performs a restrained approximation to the CSC MMSE estimator w.r.t. the entire image, explaining its superiority. To do so, we first formalize the PA approach, then we present the CSC MMSE estimator and finally, we show their connection.

PA obtains a clean estimate for each patch and averages overlapping estimates together. Specifically, under a (local) sparse prior with a dictionary $\boldsymbol{D}_L$, each patch participates in a local pursuit independently, leading to $N$ independent optimization problems,[7]

$$\left\{ \widehat{\boldsymbol{\alpha}}_i = \arg\min_{\boldsymbol{\alpha}_i} \lambda_i \|\boldsymbol{\alpha}_i\|_1 + \frac{1}{2}\|\boldsymbol{D}_L\boldsymbol{\alpha}_i - \boldsymbol{P}_i\boldsymbol{Y}\|_2^2 \right\}_{i=1}^{N}. \tag{8}$$

Once $\widehat{\boldsymbol{\alpha}}_i$ are found, the clean patches are synthesized by $\widehat{\boldsymbol{x}}_i = \boldsymbol{D}_L\widehat{\boldsymbol{\alpha}}_i$, and these are placed in the signal while averaging overlapping elements from different patches – see Equation (3).

We move now to discuss the MMSE estimation under a sparsity-promoting prior [44, 45]. Using marginalization, MMSE of the global convolutional sparse representation vector can be written as

$$\widehat{\boldsymbol{\Gamma}}_{\text{MMSE}} = \mathbb{E}\{\boldsymbol{\Gamma}|\boldsymbol{Y}\} = \mathbb{E}_S\{\mathbb{E}\{\boldsymbol{\Gamma}|\boldsymbol{Y}, S\}\} = \sum_{S \in \Theta} P(S)\,\mathbb{E}\{\boldsymbol{\Gamma}|\boldsymbol{Y}, S\} = \sum_{S \in \Theta} P(S)\,\widehat{\boldsymbol{\Gamma}}_S, \tag{9}$$

where $S$ stands for the support of $\boldsymbol{\Gamma}$, $P(S)$ is the prior probability of such a support (assumed to promote sparse vectors) and $\Theta$ is the set of all possible supports. Furthermore, $\widehat{\boldsymbol{\Gamma}}_S = \mathbb{E}\{\boldsymbol{\Gamma}|\boldsymbol{Y}, S\}$ is the MMSE estimator of the sparse representation vector given the support and the noisy measurements, known as the *oracle estimator* [44]. Equation (9) suggests that the MMSE estimator is actually a dense vector consisting of a weighted average of all the possible oracle estimators, where the weight of each is its prior probability, $P(S)$. Note that computing the MMSE is an exhaustive task that sweeps through all the possible supports, and therefore approximation methods are needed. A natural strategy in this context is to sample a sufficient number of supports from $P(S)$, and replace the expectation with a sample mean over these.

Indeed, consider the case where the sampled supports are such that $\boldsymbol{D}\boldsymbol{\Gamma}$ results in non-overlapping tangent slices. Overall, there are $n$ different slice arrangements that uphold this assumption differing only in the location of the first slice on the image. Equivalently, the $k$-th arrangement can be described by a convolutional strided dictionary, where the stride equals the size of the filter $n$, and the first non-zero needle in $\boldsymbol{\Gamma}$ is located in the $k$-th index $1 \le k \le n$. We shall denote this strided dictionary by $\boldsymbol{D}_k$ and its corresponding representation as $\boldsymbol{\Gamma}_k$. Sparse coding the $k$-th shift can be done using BP, which under these constraints can be written as

$$\widehat{\boldsymbol{\Gamma}}_k = \arg\min_{\boldsymbol{\Gamma}} \ \lambda\|\boldsymbol{\Gamma}\|_1 + \frac{1}{2}\|\boldsymbol{D}_k\boldsymbol{\Gamma} - \boldsymbol{Y}\|_2^2 \tag{10}$$

$$= \arg\min_{\boldsymbol{\Gamma}=[\boldsymbol{\alpha}_k;\boldsymbol{\alpha}_{k+n},\dots]} \sum_{i=0}^{\frac{N}{n}-1} \left\{ \lambda\|\boldsymbol{\alpha}_{k+in}\|_1 + \frac{1}{2}\|\boldsymbol{D}_L\boldsymbol{\alpha}_{k+in} - \boldsymbol{P}_{k+in}\boldsymbol{Y}\|_2^2 \right\}. \tag{11}$$

This can be solved for each needle separately,

$$\boldsymbol{\alpha}_{k+in} = \arg\min_{\boldsymbol{\alpha}} \ \lambda\|\boldsymbol{\alpha}\|_1 + \frac{1}{2}\|\boldsymbol{D}_L\boldsymbol{\alpha} - \boldsymbol{P}_{k+in}\boldsymbol{Y}\|_2^2, \tag{12}$$

while zeroing all the other needles. Thus, under the constraint of non-overlapping slices, estimating the CSC representation $\widehat{\boldsymbol{\Gamma}}_S$ is equivalent to $\frac{N}{n}$ independent local pursuits. Clearly, this estimation results with a "blockified" image, due to the lack of overlaps.

Repeating this estimation process $n$ times, each time forcing a different shift $1 \leq k \leq n$, leads to a set of estimates $\left\{ \widehat{\boldsymbol{\Gamma}}_1, \widehat{\boldsymbol{\Gamma}}_2, ..., \widehat{\boldsymbol{\Gamma}}_n \right\}$, where $\widehat{\boldsymbol{\Gamma}}_k$ denotes the estimate obtained using the $k$-th shift. Looking back into the MMSE estimator in Equation (9), the MMSE can be approximated as

$$\widehat{\boldsymbol{\Gamma}}_{\text{MMSE}} \approx \sum_{i=1}^{n} P\left(S\right) \widehat{\boldsymbol{\Gamma}}_i \approx \frac{1}{n} \sum_{i=1}^{n} \widehat{\boldsymbol{\Gamma}}_i, \tag{13}$$

if we further assume that all the estimates are a-priori equally likely. Since each $\widehat{\boldsymbol{\Gamma}}_i$ is obtained by a local non-overlapping pursuit (11), the result in (13) is exactly the above outlined PA procedure. Hence, PA can be perceived as an MMSE approximation of the CSC model, explaining its superior MSE performance when compared to a single global CSC pursuit.[8] Armed with this insight, can we propose better CSC MMSE estimates? This takes us to the next section.

## 4 The Proposed Approach

### 4.1 Generalizing the MMSE Approximation Using Strided Convolutions

We suggest to generalize the non-overlapping slices assumption and allow for a smaller constant stride. Formally, in the non-overlapping case, the stride between adjacent slices was of the same size as the filters themselves, leading to $n$ such estimates. We suggest to use a stride $q$, where $1 \leq q < n$, leading to $q$ estimates in a 1D signal or $q^2$ in a 2D one – each originating from a different initial shift in the signal. Finally, we average these together, as suggested in Equation (9). Note that when $q < n$, each estimate allows for overlapping slices, implying that the pursuit must be done globally on all the involved slices together. This necessarily leads to a global agreement between these slices, as opposed to PA where each patch (slice) is estimated separately. Furthermore, when $q$ is sufficiently large, the mutual coherence of the global dictionary can be preserved even for smooth filters, in contrast to the standard CSC pursuit ($q = 1$), since the filters only partially overlap.

For a preliminary evaluation of our approach, we perform a denoising experiment on images from the Set12 dataset contaminated with white Gaussian noise with standard deviation $\sigma \in \{15, 25, 50, 75\}$. We use both the standard PA algorithm, and the proposed strided CSC using various strides $1 \leq q < n = 11$, followed by an averaging operation. BP in its error-bounded from followed by a debiasing step is used to sparse code the signals both in the convolutional and the PA cases. The twice over-redundant DCT dictionary of size $11 \times 11$ is chosen as the local dictionary $\boldsymbol{D}_L$. Note that in the strided case, pixels may now have a different number of slices (filters) overlapping them, depending on their position in the image and the stride. To compensate for this, we normalize the filters appropriately for each stride.

A summary of the results of this experiment are presented in Table 1 (per-image results can be found in the supplementary material). As expected, when using CSC with a stride of 1, i.e. standard CSC, the denoising performance is poor and the PA method is substantially better. We attribute this to the high coherence of the global dictionary, making the estimated image overfit the noise. However, the best results are achieved when the stride is large but smaller than the size of the filters, restraining the coherence, while allowing the filters to overlap, leading to a global consensus in each of the estimates. Interestingly, the CSC achieved better results even though its error constraint ($\|\boldsymbol{D_k}\boldsymbol{\Gamma} - \boldsymbol{Y}\|_2 \leq \epsilon$) is global, as opposed to the much more detailed local constraint used by PA.

### 4.2 CSCNet – a Supervised Denoising Model

A popular method to solve BP, i.e. $\widehat{\boldsymbol{\Gamma}} = \arg\min \boldsymbol{\Gamma} \ \frac{1}{2} \|\boldsymbol{D}\boldsymbol{\Gamma} - \boldsymbol{Y}\|_2^2 + \lambda \|\boldsymbol{\Gamma}\|_1$, is the ISTA algorithm, which operates iteratively as follows:

$$\boldsymbol{\Gamma}_{k+1} = \mathcal{S}_{\frac{\lambda}{c}} \left( \boldsymbol{\Gamma}_k + \frac{1}{c} \boldsymbol{D}^T \left( \boldsymbol{Y} - \boldsymbol{D}\boldsymbol{\Gamma}_k \right) \right), \tag{14}$$

where $c \geq \sigma_{\max} \left( \boldsymbol{D}^T \boldsymbol{D} \right)$, and $\mathcal{S}_\tau$ is the soft-thresholding operator extended to operate in an element-wise fashion.[9] Often times, convergence requires a large number of iterations, making this process

Table 1: Average Set12 denoising results (PSNR) using PA and CSC with various strides ($q$). CSC Results that surpass PA are marked in blue. Best results are bold.

| | CSC - stride size ($q$) | | | | | | | | | | PA |
|---|---|---|---|---|---|---|---|---|---|---|---|
| $\sigma$ | 1 | 2 | 3 | 4 | 5 | 6 | 7 | 8 | 9 | 10 | |
| 15 | 28.99 | 29.27 | 30.01 | 30.66 | 31.06 | 31.21 | 31.31 | 31.39 | 31.45 | **31.46** | 31.23 |
| 25 | 25.78 | 26.11 | 26.94 | 27.72 | 28.26 | 28.50 | 28.64 | 28.75 | 28.84 | **28.88** | 28.73 |
| 50 | 21.49 | 22.11 | 23.17 | 23.83 | 24.52 | 24.86 | 25.05 | 25.29 | 25.47 | **25.56** | 25.32 |
| 75 | 18.83 | 19.58 | 20.95 | 21.81 | 22.43 | 22.75 | 22.97 | 23.25 | 23.51 | **23.66** | 23.28 |

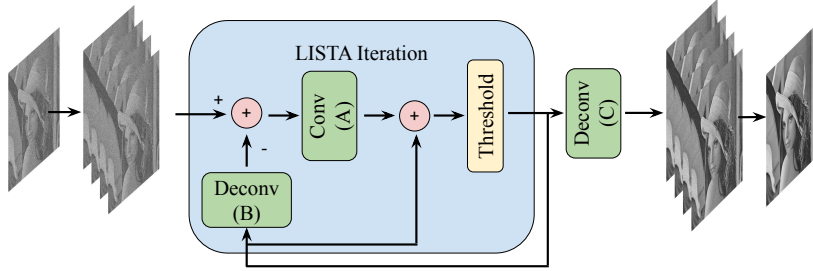

Figure 2: The CSCNET architecture.

inefficient. To overcome this burden, the LISTA algorithm [38] has been proposed to approximate the sparse coding process, by learning the parameters of a non-linear recurrent encoder that strictly follows $L$ iterations of the iterative process described in Equation (14). This concept has been extended to the convolutional setting in [37] as follows:

$$\mathbf{\Gamma}_{k+1} = \mathcal{S}_{\boldsymbol{\tau}}\left(\mathbf{\Gamma}_k + \frac{1}{c}\boldsymbol{A}\left(\boldsymbol{Y} - \boldsymbol{B}\mathbf{\Gamma}_k\right)\right), \tag{15}$$

where $\boldsymbol{A}$ stands for a convolution operator and $\boldsymbol{B}$ a transposed-convolution one. Once the sparse vector is at hand, the estimated clean image is then obtained by a linear transposed-convolutional decoder, i.e. $\widehat{\boldsymbol{X}} = \boldsymbol{C}\mathbf{\Gamma}_L$. The matrices $\boldsymbol{A}, \boldsymbol{B}$ and $\boldsymbol{C}$ are structured as a set of support bounded shift invariant filters, and together with the thresholds vector $\boldsymbol{\tau}$, are learned in a supervised manner. Note that the number of parameters does not grow with $L$, the number of unrolled iterations.

Following the CSC MMSE approximation introduced in Section 4.1, we propose to use a strided convolutional structure on the learned matrices using a constant stride $q$. To obtain an estimate for each possible shift, we duplicate the input image $q^2$ times, where each duplicate is a shifted version of the original image. Following Equation (13) the estimated image is a simple average of the estimates of all the shifts. A diagram of the proposed architecture is presented in Figure 2.

## 4.3 Experiments

To train the proposed model, we prepare a training set of input-output pairs. The clean images are taken from the Waterloo Exploration Dataset [46] and 432 images from BSD [47]. The noisy inputs are obtained by adding white Gaussian noise with a constant standard deviation $\sigma$. In each iteration, a random patch of size 128 is cropped from an image and a random realization of noise is sampled.

We train 4 models, one for each noise level $\{15, 25, 50, 75\}$. For each model we learn 175 filters of size $11 \times 11$, use a stride $q = 8$ and set $L = 12$. To learn the parameters of the model, we employ the ADAM optimizer [48] and minimize the $\ell_2$ loss, i.e. $\mathcal{L}(\boldsymbol{X}, \widehat{\boldsymbol{X}}) = \|\boldsymbol{X} - \widehat{\boldsymbol{X}}\|_2^2$. We use a learning rate of $10^{-4}$ and decrease it by a factor of 0.7 every 50 epochs and iterate over 250 epochs. To avoid divergence, we set the $\epsilon$ parameter of the optimizer to $10^{-3}$. We evaluate the performance of the models using the BSD68 dataset that was excluded from the training set. Additional experiments and information can be found in the supplementary material and on https://github.com/drorsimon/CSCNet.

Table 2 presents the results of our models compared to other leading methods, and Figure 3 shows some of the learned filters, taken from $\boldsymbol{C}$. The proposed model outperforms BM3D [2], TNRD [49],

Table 2: Denoising performance (PSNR) on the BSD68 dataset.

| $\sigma$ | BM3D | WNNM | TNRD | MLP | DnCNN | FFDNet | CSCNet |
|---|---|---|---|---|---|---|---|
| 15 | 31.07 | 31.37 | 31.42 | – | 31.72 | 31.63 | 31.57 |
| 25 | 28.57 | 28.83 | 28.92 | 28.96 | 29.22 | 29.19 | 29.11 |
| 50 | 25.62 | 25.87 | 25.97 | 26.03 | 26.23 | 26.29 | 26.24 |
| 75 | 24.21 | 24.40 | – | 24.59 | 24.64 | 24.79 | 24.77 |

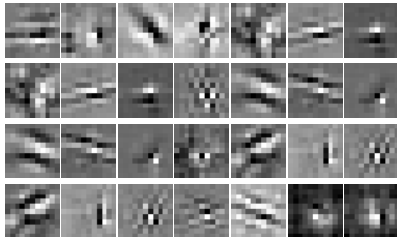

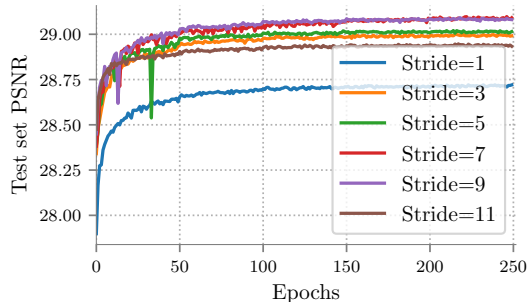

Figure 3: CSCNet filters.　　　Figure 4: CSCNet test error for various strides.

WNNM [50] and MLP [51], while being on par with DnCNN [52] and FFDNet [53]. That said, we mention two differences between the proposed model and the other two leading methods:

1. Number of parameters – The number of parameters in the proposed approach does not grow with the depth of the model. Hence, it uses much fewer parameters compared to other modern methods, as demonstrated in Table 3.

2. Batch Normalization (BN) – The other two leading denoising methods are based on general deep learning techniques, and therefore employ BN which is known to improve the performance and convergence rate of the trained model [54]. As our presented method relies only on the CSC prior, we did not include such operators.

To study the effect of the stride-size, we have trained 6 models, each one with a different stride. The results in Figure 4, referring to noise level $\sigma = 25$, show the same tendency as in Table 1, namely, setting the stride too high ($q = 11$) results in independent patch based processing with weaker performance (28.9dB); setting it to $q = 1$ leads to a regular (non-MMSE) deployment of the CSC with highly correlated atoms and the weakest estimate (28.74dB), which is still better than BM3D. The best results are obtained for $q = 7$ or $q = 8$ (29.11dB).

Table 3: Comparison of number of parameters in leading denoising architectures.

| Model | First layer | Last layer | Mid layers | Total |
|---|---|---|---|---|
| DnCNN | $3 \times 3 \times 1 \times 64$ | $3 \times 3 \times 64 \times 1$ | $(3 \times 3 \times 64 \times 64 + 128) \times 15$ | 556,032 |
| FFDNet | $3 \times 3 \times 5 \times 64$ | $3 \times 3 \times 64 \times 4$ | $(3 \times 3 \times 64 \times 64 + 128) \times 13$ | 486,080 |
| CSCNet | – | $11 \times 11 \times 175 \times 1$ | $(11 \times 11 \times 175 \times 1) \times 2 + 175$ | 63,700 |

We further test our approach on color image denoising. We perform similar experiments to those described earlier, where this time the input image and each filter have 3 channels (RGB). The denoising performance of our architecture is presented in Table 4. As before, our results are on par with other leading methods (DnCNN, FFDNet) while using much fewer parameters.

## 5 Conclusions

This work exposed the limitations of the CSC model in representing natural images in the presence of noise. Investigation of the patch-averaging scheme and the origins for its success has lead us to offer an MMSE approximation pursuit that overcomes these limitations effectively. A feed-forward architecture based on our insights was shown to be on par with the best supervised denoising

Table 4: Denoising performance (PSNR) on the color-BSD68 dataset.

| $\sigma$ | CBM3D | CDnCNN | FFDNet | CSCNet |
|---|---|---|---|---|
| 15 | 33.52 | 33.89 | 33.87 | 33.83 |
| 25 | 30.71 | 31.23 | 31.21 | 31.18 |
| 50 | 27.38 | 27.92 | 27.96 | 28.00 |
| 75 | 25.74 | 24.47 | 26.24 | 26.32 |

algorithms in the literature. Our future work will focus on further improvements of this scheme by considering (i) the addition of batch-normalization; (ii) a multi-scale architecture, as proposed in recent leading methods; (iii) adoption of a local error constraint as in PA; and (iv) exploiting self-similarity, as practiced in [20] and more recently in [55]. In all these directions, our prime goal is to incorporate these ideas while maintaining the purity of CSC model, so as to preserve intact the theoretical justification of the proposed architecture.

## Acknowledgement

The research leading to these results has received funding from the Technion Hiroshi Fujiwara Cyber Security Research Center and the Israel Cyber Directorate.

## Footnotes

[1] $\ell_0$ is not formally a norm since it does not satisfy the homogeneity property.

[2]These represent convolutions with small support filters.

[3]For simplicity of the description, and without loss of generality, we describe the CSC throughout this paper as operating on 1D signals.

[4]This is measured via an $\ell_{0,\infty}$ pseudo-norm, defined as $\|\boldsymbol{\Gamma}\|_{0,\infty} = \max_i \|\boldsymbol{\gamma}_i\|_0$.

[5]Improved performance is reported in their follow-up thesis.

[6]The $\ell_{0,\infty}$ is defined as $\|\mathbf{\Gamma}\|_{0,\infty} = \max_i \|\mathbf{\gamma}_i\|_0$.

[7] we assume the use of the BP in its Lagrangian form

[8]The term approximation refers to considering only a small subset of supports in the averaging process.

[9]$\sigma_{\max}\left(\cdot\right)$ represent the largest eigenvalue, and $\mathcal{S}_\tau\left(\cdot\right)$ is defined as $\mathcal{S}_\tau\left(y\right) = \text{sign}(y) \cdot \max(y - \tau, 0)$.

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
