[Supplementary Material]

# Rethinking the CSC Model for Natural Images
# Supplementary Material

This document contains supplementary material for "Rethinking the CSC Model for Natural Images" paper, submitted to NeurIPS 2019.

## 1  Strided CSC Results

In Table 1 we provide a per-image detailed comparison for the Set12 dataset (see Figure 1) between the PA method and the strided CSC concept that was introduced in the paper under Section 4. Note that this experiment corresponds to a fixed (not learned) twice redundant DCT dictionary.

Figure 1: The Set12 images.

## 2  CSCNet Details

### 2.1  Parameter Initialization

Since the CSCNet architecture imitates the operation of the ISTA algorithm, we initialize its parameters such that each layer will indeed follow an ISTA iteration. To do so, we first initialize a random set of filters that are similar to a CSC dictionary, i.e. a deconvolution operator $D$ and normalize it by $\sqrt{\sigma_{\max}\left(D^T D\right)}$, where the largest eigenvalue is found using the power method, implemented to operate on convolutional operators. Once the operator is normalized, we initialize the parameters of the network as follows: $A = D^T$, $B = D$ and $C = D$. For the soft thresholding operator, we set its thresholds vector $\tau$ to a constant vector with the value $0.01$, which is equivalent to setting the $\lambda$ parameter of the original Lagrangian problem to $\lambda = 0.01$ which has been empirically found to work well. In the iterative process itself, we assume that $\Gamma_0$, i.e. the initial sparse representation, is zero.

## 2.2 Training CSCNet

To train this model we use the Pytorch framework [1] installed on a standard PC containing an Intel(R) Core(TM) i7-6900K CPU and a single Nvidia GeForce GTX 1080Ti. Training a model on this setup takes approximately 14 hours. The code used to train the model, alongside the trained models that were used in this paper are publicly available at `https://anonymous.4open.science/r/a550eeb5-942e-4ffc-b75d-f0fb3d58785f/`.

## 2.3 Hyper-parameters Comparison

In this subsection we explore the effect of various parameters used in our model. Unless stated specifically otherwise, in the following described experiments the standard deviation of the noise is $\sigma = 25$, the filters used have a size of $11 \times 11$, the stride chosen is $q = 8$ and we train for 250 epochs. The BSD68 dataset is used to evaluate the performance of the various models.

The first hyper-parameter we consider is the size of the stride. The paper itself discusses the effect of the size of the stride on the trained model. As presented in the paper, a small stride leads to highly correlated filters reducing the performance of the model. On the other extreme, when the stride is too large, the performace reduces due to the lack of sufficient overlap between the filters.

The next parameter we examine is the depth of the network, i.e. the number of LISTA iterations used. Figure 2 demonstrates the effect of the depth of the network on the denoising performance. As expected, a deeper network leads to higher PSNR results. That said, the improvement seems to saturate at a certain depth. We chose a depth of 12 as a compromise between the performance of the model and its training time.

Another parameter to consider is the size of the filters. Figure 3 demonstrates the performance of various models trained with different filter sizes. From the figure we learn that increasing the size of the filters improves the denoising performance of the trained models. That said, this also increases the number of computations required to obtain an estimate, making the learning process much slower. Training a model with filter size of $15 \times 15$ took approximately two days, while improving the performance by a relatively small margin (0.06dB). Hence, in our reported results, we chose $11 \times 11$ for the size of the filters.

The last parameter we examine is the number of epochs. We trained our model with a filter size of $11 \times 11$ for 500 epochs, and demonstrate its denoising performance on the BSD68 test set during the training phase in Figure 4. From the figure, it seems that one can improve the denoising results further by continuing the training process, but only by an insignificant increase in performance. Therefore, we have decided to stop the training process after 250 epochs, practically reaching convergence.

Figure 2: The effect of the depth of the CSCNet on its denosing performance.

Figure 3: The effect of the size of the filters on the denoising performance of CSCNet.

Figure 4: The convergence of the CSCNet model over the number of epochs.

## 2.4 Additional Results

Table 2 Shows the denoising performance of our proposed method, alongside other leading denoising algorithms on the Set12 dataset. Note that our method is indeed on par with current leading methods, especially when the energy of the noise is high.

## References

[1] Adam Paszke, Sam Gross, Soumith Chintala, Gregory Chanan, Edward Yang, Zachary DeVito, Zeming Lin, Alban Desmaison, Luca Antiga, and Adam Lerer. Automatic differentiation in pytorch. 2017.

Table 1: Set12 denoising results (PSNR) using PA and CSC with various strides ($q$). CSC Results that surpass PA are marked in blue. Best results are bold.

| Image | $q=1$ | $q=2$ | $q=3$ | $q=4$ | $q=5$ | $q=6$ | $q=7$ | $q=8$ | $q=9$ | $q=10$ | PA |
|---|---|---|---|---|---|---|---|---|---|---|---|
| | | | | | CSC | | | | | | |
| $\sigma = 15$ | | | | | | | | | | | |
| C.man | 28.47 | 28.73 | 29.40 | 30.00 | 30.36 | 30.54 | 30.63 | 30.71 | 30.77 | **30.80** | 30.66 |
| House | 29.91 | 30.17 | 31.33 | 32.38 | 32.98 | 33.28 | 33.34 | 33.46 | 33.57 | **33.64** | 33.44 |
| Peppers | 28.90 | 29.14 | 29.97 | 30.66 | 31.06 | 31.20 | 31.38 | 31.45 | **31.49** | 31.43 | 31.09 |
| Starfish | 28.26 | 28.35 | 28.80 | 29.35 | 29.76 | 29.82 | 29.96 | 30.06 | 30.14 | **30.15** | 29.86 |
| Monarch | 28.43 | 28.59 | 29.20 | 29.83 | 30.28 | 30.39 | 30.53 | 30.61 | **30.64** | 30.62 | 30.29 |
| Airplane | 27.95 | 28.19 | 28.85 | 29.45 | 29.77 | 29.99 | 30.11 | 30.18 | 30.21 | **30.23** | 30.17 |
| Parrot | 28.55 | 28.76 | 29.37 | 29.93 | 30.26 | 30.45 | 30.57 | 30.63 | 30.70 | **30.73** | 30.51 |
| Lena | 30.17 | 30.76 | 31.87 | 32.66 | 33.11 | 33.24 | 33.34 | 33.46 | **33.52** | 33.50 | 33.12 |
| Barbara | 29.60 | 30.10 | 30.92 | 31.56 | 32.00 | 32.08 | 32.15 | 32.27 | **32.30** | 32.26 | 31.90 |
| Boat | 29.23 | 29.54 | 30.27 | 30.81 | 31.17 | 31.32 | 31.38 | 31.43 | 31.51 | **31.56** | 31.40 |
| Man | 29.23 | 29.49 | 30.09 | 30.61 | 30.95 | 31.06 | 31.15 | 31.22 | 31.27 | **31.31** | 31.11 |
| Couple | 29.18 | 29.37 | 30.05 | 30.63 | 30.97 | 31.11 | 31.18 | 31.23 | 31.26 | **31.30** | 31.17 |
| $\sigma = 25$ | | | | | | | | | | | |
| C.man | 25.19 | 25.45 | 26.24 | 26.99 | 27.44 | 27.75 | 27.87 | 27.98 | 28.10 | **28.18** | 28.09 |
| House | 26.46 | 27.00 | 28.18 | 29.34 | 30.07 | 30.46 | 30.57 | 30.77 | 30.97 | **31.08** | 31.01 |
| Peppers | 25.70 | 25.88 | 26.80 | 27.70 | 28.35 | 28.63 | 28.78 | 28.89 | **28.93** | 28.91 | 28.71 |
| Starfish | 25.02 | 25.14 | 25.74 | 26.38 | 26.84 | 26.98 | 27.17 | 27.26 | 27.32 | **27.37** | 27.18 |
| Monarch | 25.18 | 25.41 | 26.11 | 26.82 | 27.34 | 27.50 | 27.71 | 27.82 | **27.87** | 27.86 | 27.54 |
| Airplane | 24.72 | 24.97 | 25.72 | 26.43 | 26.89 | 27.20 | 27.34 | 27.42 | 27.47 | 27.52 | **27.59** |
| Parrot | 25.30 | 25.55 | 26.29 | 27.01 | 27.53 | 27.83 | 27.94 | 28.05 | 28.15 | **28.20** | 28.02 |
| Lena | 26.95 | 27.54 | 28.78 | 29.73 | 30.41 | 30.66 | 30.80 | 30.97 | 31.07 | **31.09** | 30.74 |
| Barbara | 26.31 | 26.83 | 27.70 | 28.46 | 29.06 | 29.28 | 29.43 | 29.58 | **29.66** | 29.64 | 29.32 |
| Boat | 26.15 | 26.51 | 27.35 | 28.03 | 28.54 | 28.72 | 28.81 | 28.89 | 28.99 | **29.07** | 29.02 |
| Man | 26.28 | 26.57 | 27.30 | 27.90 | 28.38 | 28.55 | 28.68 | 28.75 | 28.84 | **28.90** | 28.78 |
| Couple | 26.18 | 26.39 | 27.11 | 27.79 | 28.27 | 28.46 | 28.54 | 28.62 | 28.68 | **28.78** | 28.73 |
| $\sigma = 50$ | | | | | | | | | | | |
| C.man | 20.83 | 21.00 | 21.98 | 22.85 | 23.63 | 24.13 | 24.34 | 24.53 | 24.72 | **24.88** | 24.80 |
| House | 21.83 | 23.18 | 24.59 | 25.43 | 26.33 | 26.61 | 26.83 | 27.26 | 27.57 | **27.78** | 27.75 |
| Peppers | 21.45 | 22.18 | 23.25 | 23.81 | 24.48 | 24.87 | 25.06 | 25.26 | 25.37 | **25.38** | 25.18 |
| Starfish | 20.83 | 21.36 | 22.21 | 22.52 | 23.10 | 23.44 | 23.74 | 23.95 | **24.07** | **24.07** | 23.73 |
| Monarch | 20.73 | 20.99 | 21.90 | 22.57 | 23.33 | 23.62 | 23.83 | 24.03 | 24.20 | **24.22** | 23.82 |
| Airplane | 20.60 | 21.45 | 22.19 | 22.68 | 23.30 | 23.71 | 23.94 | 24.09 | 24.18 | **24.30** | 24.23 |
| Parrot | 20.91 | 21.14 | 22.13 | 22.96 | 23.66 | 24.07 | 24.28 | 24.52 | 24.71 | **24.81** | 24.29 |
| Lena | 22.48 | 23.40 | 24.87 | 25.76 | 26.53 | 26.91 | 27.09 | 27.41 | 27.65 | **27.75** | 27.47 |
| Barbara | 21.87 | 22.41 | 23.48 | 24.31 | 24.96 | 25.35 | 25.53 | 25.77 | 25.98 | **26.06** | 25.61 |
| Boat | 22.03 | 22.56 | 23.61 | 24.30 | 24.96 | 25.27 | 25.40 | 25.61 | 25.80 | **25.93** | 25.81 |
| Man | 22.23 | 22.93 | 24.15 | 24.75 | 25.25 | 25.44 | 25.60 | 25.80 | 25.97 | **26.05** | 25.84 |
| Couple | 22.13 | 22.71 | 23.65 | 24.06 | 24.67 | 24.93 | 25.02 | 25.23 | 25.40 | **25.54** | 25.36 |
| $\sigma = 75$ | | | | | | | | | | | |
| C.man | 18.26 | 19.00 | 20.05 | 20.67 | 21.35 | 21.96 | 22.28 | 22.56 | 22.81 | **23.00** | 22.95 |
| House | 18.88 | 20.17 | 22.08 | 23.25 | 24.06 | 24.38 | 24.65 | 25.08 | 25.45 | **25.69** | 25.57 |
| Peppers | 18.66 | 19.51 | 21.09 | 21.89 | 22.43 | 22.64 | 22.83 | 23.04 | 23.22 | **23.27** | 22.98 |
| Starfish | 18.26 | 18.96 | 20.28 | 20.94 | 21.39 | 21.45 | 21.78 | 22.00 | 22.23 | **22.30** | 21.73 |
| Monarch | 18.13 | 18.65 | 19.91 | 20.60 | 21.23 | 21.53 | 21.83 | 22.16 | 22.34 | **22.37** | 21.85 |
| Airplane | 18.08 | 19.34 | 20.59 | 21.10 | 21.48 | 21.65 | 21.81 | 22.02 | 22.19 | **22.28** | 22.07 |
| Parrot | 18.25 | 18.70 | 20.03 | 20.85 | 21.43 | 21.85 | 22.14 | 22.42 | 22.67 | **22.81** | 22.43 |
| Lena | 19.72 | 20.51 | 22.21 | 23.46 | 24.26 | 24.71 | 24.87 | 25.25 | 25.64 | **25.86** | 25.33 |
| Barbara | 19.20 | 19.66 | 20.75 | 21.74 | 22.49 | 23.00 | 23.13 | 23.39 | 23.69 | **23.90** | 23.25 |
| Boat | 19.41 | 20.03 | 21.38 | 22.25 | 22.89 | 23.22 | 23.40 | 23.67 | 23.96 | **24.17** | 23.83 |
| Man | 19.61 | 20.26 | 21.63 | 22.68 | 23.37 | 23.66 | 23.80 | 24.08 | 24.37 | **24.54** | 23.88 |
| Couple | 19.47 | 20.17 | 21.45 | 22.27 | 22.78 | 22.96 | 23.10 | 23.31 | 23.56 | **23.74** | 23.48 |

Table 2: Set12 denoising results (PSNR) comparison using various methods. The best and second-best results are highlighted in red and blue respectively.

| Image | BM3D | WNNM | MLP | TNRD | DnCNN | FFDNet | CSCNet |
|-------|------|------|-----|------|-------|--------|--------|
| $\sigma = 15$ | | | | | | | |
| C.man | 31.91 | 32.17 | – | 32.19 | 32.61 | 32.42 | 32.40 |
| House | 34.93 | 35.13 | – | 34.53 | 34.97 | 35.01 | 34.96 |
| Peppers | 32.69 | 32.99 | – | 33.04 | 33.30 | 33.10 | 33.19 |
| Starfish | 31.14 | 31.82 | – | 31.75 | 32.20 | 32.02 | 31.89 |
| Monarch | 31.85 | 32.71 | – | 32.56 | 33.09 | 32.77 | 32.78 |
| Airplane | 31.07 | 31.39 | – | 31.46 | 31.70 | 31.58 | 31.60 |
| Parrot | 31.37 | 31.62 | – | 31.63 | 31.83 | 31.77 | 31.86 |
| Lena | 34.26 | 34.27 | – | 34.24 | 34.62 | 34.63 | 34.44 |
| Barbara | 33.10 | 33.60 | – | 32.13 | 32.64 | 32.50 | 32.22 |
| Boat | 32.13 | 32.27 | – | 32.14 | 32.42 | 32.35 | 32.29 |
| Man | 31.92 | 32.11 | – | 32.23 | 32.46 | 32.40 | 32.31 |
| Couple | 32.10 | 32.17 | – | 32.11 | 32.47 | 32.45 | 32.31 |
| Average | 32.37 | 32.70 | – | 32.50 | 32.86 | 32.75 | 32.69 |
| $\sigma = 25$ | | | | | | | |
| C.man | 29.45 | 29.64 | 29.61 | 29.72 | 30.18 | 30.06 | 29.95 |
| House | 32.85 | 33.22 | 32.56 | 32.53 | 33.06 | 33.27 | 33.07 |
| Peppers | 30.16 | 30.42 | 30.30 | 30.57 | 30.87 | 30.79 | 30.74 |
| Starfish | 28.56 | 29.03 | 28.82 | 29.02 | 29.41 | 29.33 | 29.02 |
| Monarch | 29.25 | 29.84 | 29.61 | 29.85 | 30.28 | 30.14 | 30.07 |
| Airplane | 28.42 | 28.69 | 28.82 | 28.88 | 29.13 | 29.05 | 28.98 |
| Parrot | 28.93 | 29.15 | 29.25 | 29.18 | 29.43 | 29.43 | 29.43 |
| Lena | 32.07 | 32.24 | 32.25 | 32.00 | 32.44 | 32.59 | 32.34 |
| Barbara | 30.71 | 31.24 | 29.54 | 29.41 | 30.00 | 29.98 | 29.36 |
| Boat | 29.90 | 30.03 | 29.97 | 29.91 | 30.21 | 30.23 | 30.11 |
| Man | 29.61 | 29.76 | 29.88 | 29.87 | 30.1 | 30.1 | 29.99 |
| Couple | 29.71 | 29.82 | 29.73 | 29.71 | 30.12 | 30.18 | 30.01 |
| Average | 29.97 | 30.26 | 30.03 | 30.06 | 30.43 | 30.43 | 30.26 |
| $\sigma = 50$ | | | | | | | |
| C.man | 26.13 | 26.45 | 26.37 | 26.62 | 27.03 | 27.03 | 26.75 |
| House | 29.69 | 30.33 | 29.64 | 29.48 | 30.00 | 30.43 | 30.25 |
| Peppers | 26.68 | 26.95 | 26.68 | 27.10 | 27.32 | 27.43 | 27.38 |
| Starfish | 25.04 | 25.44 | 25.43 | 25.42 | 25.70 | 25.77 | 25.56 |
| Monarch | 25.82 | 26.32 | 26.26 | 26.31 | 26.78 | 26.88 | 26.54 |
| Airplane | 25.10 | 25.42 | 25.56 | 25.59 | 25.87 | 25.90 | 25.86 |
| Parrot | 25.90 | 26.14 | 26.12 | 26.16 | 26.48 | 26.58 | 26.55 |
| Lena | 29.05 | 29.25 | 29.32 | 28.93 | 29.39 | 29.68 | 29.53 |
| Barbara | 27.22 | 27.79 | 25.24 | 25.70 | 26.22 | 26.48 | 25.72 |
| Boat | 26.78 | 26.97 | 27.03 | 26.94 | 27.20 | 27.32 | 27.24 |
| Man | 26.81 | 26.94 | 27.06 | 26.98 | 27.24 | 27.30 | 27.15 |
| Couple | 26.46 | 26.64 | 26.67 | 26.50 | 26.90 | 27.07 | 27.07 |
| Average | 26.72 | 27.05 | 26.78 | 26.81 | 27.18 | 27.32 | 27.13 |
| $\sigma = 75$ | | | | | | | |
| C.man | 24.32 | 24.6 | 24.63 | – | 25.07 | 25.29 | 25.15 |
| House | 27.51 | 28.24 | 27.78 | – | 27.85 | 28.43 | 28.70 |
| Peppers | 24.73 | 24.96 | 24.88 | – | 25.17 | 25.39 | 25.53 |
| Starfish | 23.27 | 23.49 | 23.57 | – | 23.64 | 23.82 | 23.55 |
| Monarch | 23.91 | 24.31 | 24.40 | – | 24.71 | 24.99 | 24.67 |
| Airplane | 23.48 | 23.74 | 23.87 | – | 24.03 | 24.18 | 24.27 |
| Parrot | 24.18 | 24.43 | 24.55 | – | 24.71 | 24.94 | 24.80 |
| Lena | 27.25 | 27.54 | 27.68 | – | 27.54 | 27.97 | 27.93 |
| Barbara | 25.12 | 25.81 | 23.39 | – | 23.63 | 24.24 | 23.75 |
| Boat | 25.12 | 25.29 | 25.44 | – | 25.47 | 25.64 | 25.55 |
| Man | 25.32 | 25.42 | 25.59 | – | 25.64 | 25.75 | 25.69 |
| Couple | 24.70 | 24.86 | 25.02 | – | 24.97 | 25.29 | 25.29 |
| Average | 24.91 | 25.23 | 25.07 | – | 25.2 | 25.49 | 25.41 |