[Reviews · NeurIPS 2019]

Reviewer 1



Overall, the paper is well written and self-contained, providing a nice introduction to SC and CSC. References are also appropriate. The topic of sparse representations for image denoising is relevant. Despite these good aspects, the novelty of the paper is very marginal and the main claim that connects PA SC with MMSE estimation with sparse priors is vague and not well justified. The network architecture proposed does not differ from previous LISTA-style methods and the only difference is essentially in the way you input the data. Specific comments: - In the paper it is argued that pre-processing is bad. It is not clear what this refers to. The authors seem to criticize mean (or smooth component) subtraction before sparse coding. Mean needs to be subtracted for sparse coding to work as also argued in the paper. Furthermore, in the experimental section there is a mention to debiasing the signals. Does this refer to mean subtraction? That contradicts the argument in the paper that SC shouldn't need pre-processing (i.e., smooth component removal). Also, it is not clear whether normalization is applied to the patches before sparse coding. This can be an advantage with respect to the convolutional model since it gives robustness to scale variation. - What is the double == in (6)? - When you extract every patch in an image and do sparse coding, that is essentially equivalent to the convolutional case. The difference comes in the way you combine the patches back (patch extraction operator and its adjoint). What the authors propose by using strided convolutions is effectively trying to do patch averaging with partial overlap. The results only marginally improve PA which could also be related to the way boundaries are handled. - How do you choose the sparsity level, the noise level is the same? Then global is better due to averaging in high dimensions. The opposite is claimed in the paper. - [37] already proposed a LISTA version of the CSC model. What is the difference? It seems the only difference is the "multi-channel" decomposition of the image. - A is convolution, B deconvolution??? - How is the duplicated image a shifted version of the original one? - I don’t see the point of mentioning batch normalization as possible improvement and not having tried it. There is no guarantee that that will be the case. - The experimental section needs further clarification in terms of number of atoms used in both cases, pre-processing steps, and algorithmic solution (e.g., Lagrangian vs L1 minimization).

Reviewer 2



I have no further comments on this paper. I think this is a very good paper.

Reviewer 3



The paper discusses two important lines of works that appeared ten years ago and have become ubiquitous in inverse problems. On one side, the dictionary learning strategy, based on patch sparse coding and then averaging. On the other side, the CSC which is based on convolution filters. A unified presentation of both worlds allows authors to explain the limits of both techniques, and to propose a new CNN with improved performance. The paper is well written and convincing, and I have only few comments: * in equation (7), I did not understand the (0,infinity) norm notation. I would say it means the level of group-sparsity, but this does not match the description "local non-zero elements". Please clarify this by providing the true definition. * The references should be polished (capital letters are often missing in the titles) * line 32: "cardinality" is often referred to as "sparsity level" * line 64: "show" -> shown

[Author Response · NeurIPS 2019]

# Rethinking the CSC Model for Natural Images – Rebuttal

We thank the reviewers for their constructive comments and their valuable input. We provide our answers below.

**Response to reviewer #1:**

**1. Patch averaging and CSC** – "*When you extract every patch in an image and do sparse coding, that is essentially equivalent to the convolutional case. The difference comes in the way you combine the patches back... using strided convolutions is effectively trying to do patch averaging with partial overlap.*" – Sparse coding of overlapping patches is very different from the convolutional model (strided or not). The main difference is that the former solves a set of independent SC optimization problems (on patches), whereas the convolutional one solves a single global one (for the entire image). Hence, these models constitute completely different methodologies, and this is key in understanding our work. If indeed the two were the same, it would have been reflected in their performance. However, from the experimental section, the two differ significantly. In our work, we show that PA is indeed connected to the CSC model, being an MMSE approximation for $q = n$, implying an averaging of $n^2$ solutions of $n$-strided CSC problems.

**2. Preprocessing** – "*In the paper it is argued that pre-processing is bad... The authors seem to criticize mean (or smooth component) subtraction before sparse coding*" – In our work we highlight the fact that previous successful use of the CSC model apply it only on the high frequencies of the image. To date, however, no clear reasoning for this step was given. In this work, one of our contributions is in providing a clear theoretical explanation to this step, which is the problem of high coherence of the global dictionary. Furthermore, we show that the best results can be obtained even without any preprocessing steps by aiming for an MMSE estimation while using large strides. In our experiments, no preprocessing nor normalization steps are applied in either patch-based or the convolutional settings.
"*...in the experimental section there is a mention to debiasing the signals...* " – Debiasing refers to a projection of the input signal onto the atoms corresponding to the found support of the sparse vector by Least-Squares. This is commonly used when solving an $\ell_1$ relaxation to the $\ell_0$ problem, and has nothing to do with the preprocessing discussed above.

**3. Relation to [37]** – "*The network architecture proposed does not differ from previous LISTA-style methods and the only difference is essentially in the way you input the data.*" – The main difference between the two architectures is that [37] originates from MAP estimation, whereas ours is inspired by the MMSE one. Furthermore, our architecture deals with the problem of high coherence by operating in a strided fashion. When the stride is $q = 1$ our model coincides with [37], and when it is $q = n$ our architecture coincides with patch averaging. However, when $1 < q < n$, we obtain a set of $q^2$ global image estimates which are finally averaged together as suggested by the formulation of the MMSE estimator (Eq. (13) in the paper). Most importantly, our scheme arises from a deep understanding of the CSC model and its limitations as were discussed throughout our work, and indeed this leads to a significant $0.4$dB improvement over [37] (in fact, using $q = 1$ leads to the worst performance).

**4. Sparsity level** – "*How do you choose the sparsity level?*" – For the experiments described in section 4.1 we use the $\ell_1$ (Basis Pursuit) in its error-bounded form, i.e. for the patch-based case we solve: $\min_\alpha \|\alpha\|_1$ s.t. $\|P_i X - D_L \alpha\| \le \epsilon$. We have added a comment regarding this matter in Section 4.1 in the paper. As we move to the learned network in section 4.2, the structure is induced from the ISTA algorithm and sparsity is governed by the supervised training.

**5. Novelty** – "*the novelty of the paper is very marginal and the main claim that connects PA SC with MMSE estimation with sparse priors is vague and not well justified.*" – We refer the reviewer to comments 1 and 2 made by reviewer #2.

**Response to reviewer #3:**

**1. The relevance of the CSC model** – "*CSC is now a quite old model, and there have been numerous works since that proposed CNNs for various image restoration tasks.*" – While the CSC model is indeed not new, it still obtains state of the art performance in several image processing applications, as was mentioned in Section 2 in our paper. We believe that our work presents another success story for the CSC model while also shedding light on its limitations regarding natural images in the non-strided case, making it relevant alongside the existence of newer alternatives.

**2. The $\ell_{0,\infty}$** – "*I did not understand the (0,infinity) norm notation*" – This notation is taken from [31], and the definition is given in a footnote on page 3. Following this comment, however, we have added an additional definition on page 4.

**3. Comparison to previous work** – "*I am not sure whether the considered supervised baselines [52] and [53] achieve the best performance in the experimental setting chosen by the authors. [52] for instance focuses on non-Gaussian noise.*" – Both these papers deal with Gaussian denoising. Indeed, [52] suggests using the same network for numerous tasks, such as single image super resolution. However, Gaussian denoising is the main application mentioned in both.

**4. Additional Experiments** – "*Conversely, it would be of interest to strengthen the numerical experiments...*" – Following this comment, we have successfully trained our architecture for color image Gaussian denoising, achieving similar results to the ones reported in [52]-[53], and added them to the paper. Furthermore, we are currently working on incorporating JPEG deblocking results, similar to those reported in [52]. Thank you for improving our paper.

[Meta-Review · NeurIPS 2019]

This paper presents a unifying framework for understanding both patch-based sparse dictionary learning and covolutional sparse coding. Both are well-studied methods, but there was previously limited understanding of the relationship between them. This paper presents a new perspective on old methods that leads to new insights and helps the authors make adjustments to the convolutional framework to improve its performance.